# Theory of resonantly enhanced photo-induced superconductivity

Christian J. Eckhardt [1,2], Sambuddha Chattopadhyay[3], Dante M. Kennes[1,2], Eugene A. Demler[4], Michael A. Sentef [1,5,6] & Marios H. Michael [1] ✉

Optical driving of materials has emerged as a versatile tool to control their properties, with photo-induced superconductivity being among the most fascinating examples. In this work, we show that light or lattice vibrations coupled to an electronic interband transition naturally give rise to electron-electron attraction that may be enhanced when the underlying boson is driven into a non-thermal state. We find this phenomenon to be resonantly amplified when tuning the boson's frequency close to the energy difference between the two electronic bands. This result offers a simple microscopic mechanism for photo-induced superconductivity and provides a recipe for designing new platforms in which light-induced superconductivity can be realized. We discuss two-dimensional heterostructures as a potential test ground for light-induced superconductivity concretely proposing a setup consisting of a graphene-hBN-SrTiO$_3$ heterostructure, for which we estimate a superconducting $T_c$ that may be achieved upon driving the system.

Engineering novel properties or realizing new phases of matter by irradiating materials with light is one of the most tantalizing prospects of modern condensed matter physics[1–3]. Laser light has been shown to be a versatile tool in a variety of systems, capable of photo-inducing ferroelectricity[4,5], switching between charge density wave states[6–8] and even optically-stabilizing ferromagnetism[9]. Arguably, one of the most exciting prospects is to use light to engineer high-temperature superconductivity. Experimentally, evidence for creating superconducting-like states in K$_3$C$_{60}$[10–12] and certain organic compounds[13] through laser driving in the THz range have made this prospect all the more tangible.

A considerable amount of theoretical effort has been directed at understanding photo-driven states, with a variety of proposals attempting to explain the phenomenology of photo-induced superconductivity[14–33]. However, a simple microscopic and experimentally realistic mechanism that predicts photo-controlled superconductivity, able to direct future experimental explorations, remains elusive. In this paper, we explore such a mechanism based on a driven boson (e.g., a phonon, photon or surface plasmon) locally coupled to an inter-band electronic transition as shown in Fig. 1. We demonstrate that these ingredients lead to a boson-mediated electron attraction that not only increases during pumping but also is resonantly amplified when the boson frequency is close to the inter-band transition energy.

In the context of equilibrium superconductivity in doped SrTiO$_3$[34–37], the importance of local interband-phonon coupling has previously been discussed. In this paper, however, we highlight the nonequilibrium properties of this model that to the best of the authors' knowledge have not been explored to date in the context of driven superconductivity. This can potentially elucidate the microscopic mechanism behind photo-induced superconductivity. Furthermore, this model offers a simple, microscopic prescription for finding new nonthermal pathways to superconductivity.

In the following, we present a generic model of a boson coupled to the interband transition of two electronic bands and investigate its properties, focusing on driven superconductivity. We demonstrate

[1]Max Planck Institute for the Structure and Dynamics of Matter, Center for Free-Electron Laser Science (CFEL), Luruper Chaussee 149, 22761 Hamburg, Germany. [2]Institut für Theorie der Statistischen Physik, RWTH Aachen University and JARA-Fundamentals of Future Information Technology, 52056 Aachen, Germany. [3]Lyman Laboratory, Department of Physics, Harvard University, Cambridge, MA 02138, USA. [4]Institute for Theoretical Physics, ETH Zürich, 8093 Zürich, Switzerland. [5]Institute for Theoretical Physics and Bremen Center for Computational Materials Science, University of Bremen, 28359 Bremen, Germany. [6]H H Wills Physics Laboratory, University of Bristol, Bristol BS8 1TL, UK. ✉e-mail: marios.michael@mpsd.mpg.de

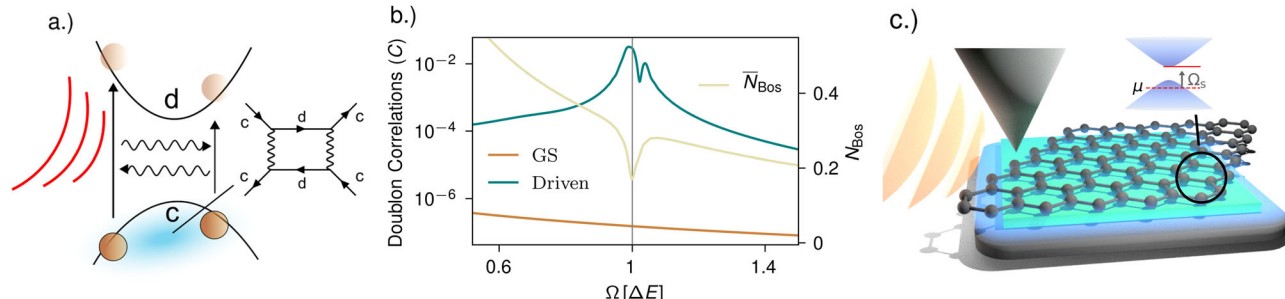

**Fig. 1 | Inter-band mechanism for photo-induced superconductivity. a** A boson coupling to an electronic inter-band transition can induce an effective electron-electron attraction stemming from virtual processes that involve the exchange of two bosons. Away from the ground state, when real bosons populate the mode, this attraction may be resonantly enhanced when tuning the boson frequency close to the transition energy. **b** Exact diagonalisation results for the proposed mechanism on two sites. Local doublon correlation C (Eq. (8)) in the two-site model given by Eq. (3) as a function of bosonic frequency $\Omega$ in units of the level separation $\Delta E$. The value in the ground state (orange line, left axis) shows no resonant behavior, but for coherently driven bosons (blue line, left axis) the time-averaged doublon correlations show a sharp increase when $\Omega \approx \Delta E$. The time-averaged boson number $\overline{N}_{\text{bos}}$ (yellow line, right axis) is also shown. The level coupling and the hopping have been set to $g = t = 0.01\Delta E$. **c** Illustration of the hetero structure we propose. Graphene (black) aligned with hBN (green) is placed on top of bulk SrTiO$_3$ (black bottom layer). Surface phonon polaritons (light blue shade) on the surface of SrTiO$_3$ couple the two graphene bands that develop a gap due to the alignment with hBN. The chemical potential $\mu$ is set such that the polariton frequency is red detuned but close to resonance. The surface polaritons can be driven by irradiating a tip (orange waves onto tip) and utilizing its near field.

that driving bosons enhances electron–electron attraction, particularly when the boson frequency matches the electronic band gap, resulting in a resonant enhancement. To support this claim we employ both perturbative methods and exact diagonalization for a two-site model, confirming the significant enhancement of electron–electron attraction on resonance. We then explore the consequences of this mechanism on superconductivity in extended systems by deriving the gap equation for our mechanism using many-body field theory. Consistent with the two site model results, we find that the resonance is absent in equilibrium, but the boson-mediated electron-electron attraction can be resonantly enhanced when the bosons are driven into a nonthermal state.

Finally we put forward two-dimensional heterostructures as a testbed for our theory. In particular, we propose to couple two-dimensional materials to the surface phonon polaritons of a substrate. Previous materials, in which light-induced superconductivity has been observed like cuprates[38], $\kappa$-salts[13] and K$_3$C$_{60}$[10–12], are strongly correlated hindering the formulation of simple theories as well as providing limited experimental control handles, making the verification of proposed mechanisms difficult. In contrast, two-dimensional (van der Waals) materials as well as surface phonon polaritons offer precise control potentially enabling direct tests of concrete microscopic theory proposals for light-induced superconductivity. As a concrete and practically realizable example, we explore a heterostructure consisting of graphene aligned with hexagonal boron nitride (hBN)[39–42] on top of the surface of SrTiO$_3$. Graphene aligned with hBN hosts two electronic bands with a small gap[43], while the SrTiO$_3$ surface provides surface phonon polaritons at a similar frequency as the gap[44,45]. We estimate that photo-induced pairing in graphene could be found up to a critical temperature of around 15 K.

## Results
### Model for photo-induced superconductivity
The generic model we propose for resonantly enhanced photo-induced superconductivity, has as basic ingredients two dispersive electronic bands:

$$H_0 = \sum_{k,\sigma} \left( \varepsilon_c(k) c^{\dagger}_{k,\sigma} c_{k,\sigma} + \varepsilon_d(k) d^{\dagger}_{k,\sigma} d_{k,\sigma} \right), \quad (1)$$

where $c^{\dagger}_{k,\sigma}$ creates an electron with momentum $k$ and spin $\sigma$ in the semi-filled conduction band with dispersion $\varepsilon_c(k)$, while $d^{\dagger}_{k,\sigma}$ creates an

electron with momentum $k$ and spin $\sigma$ in a higher lying conduction band with dispersion $\varepsilon_d(k) > \varepsilon_c(k)$. Photo-induced superconductivity manifests when a boson is coupled to the interband transition via the Hamiltonian:

$$H = H_0 + H_{\text{int}} + \sum_q \Omega(q) b^{\dagger}_q b_q,$$
$$H_{\text{int}} = \sum_{q,k,\sigma} g(q) X_q \left( c^{\dagger}_{k+q,\sigma} d_{k,\sigma} + h.c. \right), \quad (2)$$

where $b_q$ annihilates; $b^{\dagger}_q$ creates a boson with momentum $q$ and frequency $\Omega(q)$ while $g(q)$ parametrizes the coupling strength to the electrons, and $X_q = b^{\dagger}_{-q} + b_q$ is proportional to the coordinate of the oscillator.

### Local approximation
To gain intuition for the microscopic mechanism that we propose, we first interrogate a local model consisting of lattice electrons where each site has two orbitals that are coupled locally via a boson.

$$H^{\text{loc}} = H_0 + H_{\text{int}}$$
$$H_0 = \sum_{\sigma,j} \frac{\Delta E}{2} \left( n^d_{j,\sigma} - n^c_{j,\sigma} \right) + \Omega b^{\dagger}_j b_j \quad (3)$$
$$H_{\text{int}} = g \sum_{j,\sigma} X_j \left( d^{\dagger}_{j,\sigma} c_{j,\sigma} + c^{\dagger}_{j,\sigma} d_{j,\sigma} \right).$$

Here $c_{j,\sigma}$ and $c^{\dagger}_{j,\sigma}$ are annihilation and creation operators of electrons in the lower level with spin $\sigma$ at site $j$ while $d_{j,\sigma}$ and $d^{\dagger}_{j,\sigma}$ are annihilation and creation operators of electrons in the upper level that is separated from the lower one by the energy gap $\Delta E$. The local density operators for the two bands are defined as $n^c_{j,\sigma} = c^{\dagger}_{j,\sigma} c_{j,\sigma}$ and $n^d_{j,\sigma} = d^{\dagger}_{j,\sigma} d_{j,\sigma}$. $b^{\dagger}_j$ and $b_j$ are bosonic creators and annihilators of the bosonic mode at site $j$ that has eigenfrequency $\Omega$ and couples the two electronic levels with coupling strength $g$ through the operator $X_j = b^{\dagger}_j + b_j$. At this point we do not consider tunneling between the two sites.

### Ground state attraction in the local model
We focus on the most simple two-site version, $j \in \{1, 2\}$, of the model in equation (3) and consider a half filled lower level with two electrons in the system overall. At vanishing coupling $g = 0$ the ground state of the system with no bosons is degenerate between two unpaired electrons

on different sites and a singlet on the same site ($E_{\uparrow,\downarrow}^{GS} = E_{\uparrow,\downarrow,-}^{GS}$). Upon turning on the coupling $g > 0$ this degeneracy is lifted: The state with a doubly occupied site obtains a slightly lower energy than two singly occupied sites which, to leading order in $g$ reads

$$E_{\uparrow,\downarrow,-}^{GS} - E_{\uparrow,\downarrow}^{GS} = -2g^4 \frac{\Omega + 2\Delta E}{(\Delta E)\Omega(\Delta E + \Omega)^2}. \tag{4}$$

We interpret this energy as an attractive interaction between the two electrons. Such an interaction has previously been noted in ref. 34.

### Attraction out of equilibrium

To account perturbatively for the out of equilibrium behavior of the model, we perform a Schrieffer–Wolf transformation of the Hamiltonian in Eq. (3) for an arbitrary number of lattice sites to eliminate the coupling to leading order in $g$. As shown in the Methods section we find

$$\begin{aligned} H^{SW} = H_0 &- g^2 \frac{\Delta E}{(\Delta E)^2 - \Omega^2} \sum_{j,\sigma} X_j^2 \left(n_{j,\sigma}^d - n_{j,\sigma}^c\right) \\ &- g^2 \frac{\Omega}{(\Delta E)^2 - \Omega^2} \left(\sum_{j,\sigma} d_{j,\sigma}^\dagger c_{j,\sigma} + c_{j,\sigma}^\dagger d_{j,\sigma}\right)^2. \end{aligned} \tag{5}$$

The occurring denominators $(\Delta E^2 - \Omega^2)^{-1}$ in the prefactors indicate resonant behavior when $\Delta E \approx \Omega$. We note that the resonance between the last two terms exactly cancels in the ground state, which can be seen by normal ordering the operators. We further analyze the second term Eq. (5) which constitutes an electronic density coupled to the squared boson displacement[25]. In ref. 14 such an interaction was considered on symmetry grounds and it was shown that it gives rise to a boson-number dependent attraction:

$$H_{att} = -4g^4 \frac{\Delta E^2}{\Omega\left((\Delta E)^2 - \Omega^2\right)^2} \sum_j \left(b_j^\dagger b_j c_{j,\uparrow}^\dagger c_{j,\downarrow}^\dagger c_{j,\downarrow} c_{j,\uparrow}\right). \tag{6}$$

Equation (6) shows that driving the bosons out of equilibrium will enhance this attraction transiently leading to a resonantly enhanced interaction. We note here that the resonance coming from the second and third term in Eq. (5), cancels in the ground state which is why the resonance is absent at equilibrium. Exactly at the resonance, $\Delta E \approx \Omega$, perturbation theory breaks down and we turn to exact numerical methods to investigate the resonance.

To explore the induced attraction away from the ground state, we perform exact diagonalisation calculations on the two-site model of Eq. (3) including a small hopping term parametrized by the hopping amplitude $t$

$$H = H^{loc} - t \sum_{j,\sigma} \left(c_{j+1,\sigma}^\dagger c_{j,\sigma} + d_{j+1,\sigma}^\dagger d_{j,\sigma}\right) + h.c. \tag{7}$$

First we compute the lower level doublon correlations in the ground state,

$$C = \sum_j \langle n_{j,\uparrow}^c n_{j,\downarrow}^c \rangle - \langle n_{j,\uparrow}^c \rangle \langle n_{j,\downarrow}^c \rangle, \tag{8}$$

as an indicator of an induced electron–electron interaction. The result is shown in Fig. 1b. Consistent with our perturbative analysis in equation (4), in the ground state, we find positive correlations indicating an effective attraction that increases towards lower frequencies but show no specific features at $\Delta E \approx \Omega$.

To analyze the out of equilibrium case, we coherently drive the bosons by adding a time-dependent term to the Hamiltonian

according to

$$H(t) = H + F(t)\sin(\omega_D t) \sum_j \left(b_j^\dagger + b_j\right). \tag{9}$$

Here $F(t)$ is a Gaussian envelope and $\omega_D$ the driving frequency that we always set resonant with the eigenfrequency of the bosons $\Omega = \omega_D$. We compute the time-averaged density correlations Eq. (8) as well as the time-averaged boson number $\int_{t_0}^{t_1} dt \langle b_j^\dagger b_j \rangle(t) = \overline{N}_{bos}$, where $t_0$ is a time shortly after the driving pulse and $t_1$ a later time after many driving periods. Details on how we perform the time evolution can be found in the methods section. We find an overall amplification of the electron–electron attraction compared to the equilibrium result. Moreover, while in equilibrium no enhancement of the interaction close to resonance $\Omega \approx \Delta E$ was found, such an enhancement can indeed be accessed out of equilibrium as can be seen in Fig. 1b. This matches the intuition based on Eq. (6). The resonance is also evident in the time-averaged number of bosons, which has a dip indicating that electrons are being excited into the upper level by absorbing bosons which would be detrimental to pairing. For completeness we note that for the two site model we observe a peak splitting at the resonance. The reason is that a finite hopping and coupling $g$ leads to level splitting potentially enabling more than one resonance. Such features are expected to be smeared out in extended systems or due to dissipation.

### Dynamic gap equation

We show that the resonantly enhanced photo-induced attraction persists in extended systems by using many-body field theory techniques. Starting from the Hamiltonian presented in Eqs. (1) and (2), we integrate out the bosons exactly using the path-integral approach yielding a dynamic electron-electron interaction. Then we perform a saddle point approximation to derive BCS type equations for the equilibrium phonon mediated superconductivity, and then discuss how they can be extended to the nonequilibrium case.

Within saddle point approximation, the intra-band pairing fields $\Delta_c = \langle \psi_{k,\uparrow,c} \psi_{k,\downarrow,c} \rangle$ and $\Delta_d = \langle \psi_{k,\uparrow,d} \psi_{k,\downarrow,d} \rangle$ are found self-consistently through two coupled gap equations shown diagrammatically in Fig. 2a:

$$\begin{aligned} \Delta_c(k,\omega) = \frac{1}{N\beta} \sum_{k',\omega'} &\frac{g(k-k')g(k'-k)\Omega(k-k')}{(\omega-\omega')^2 + \Omega(k-k')^2} \\ &\frac{\Delta_d(k',\omega')}{2E_c(k')} \left(\frac{1}{i\omega' - E_c(k')} - \frac{1}{i\omega' + E_c(k')}\right), \end{aligned} \tag{10}$$

$$\begin{aligned} \Delta_d(k,\omega) = \frac{1}{N\beta} \sum_{k',\omega'} &\frac{g(k-k')g(k'-k)\Omega(k-k')}{(\omega-\omega')^2 + \Omega(k-k')^2} \\ &\frac{\Delta_c(k',\omega')}{2E_d(k')} \left(\frac{1}{i\omega' - E_d(k')} - \frac{1}{i\omega' + E_d(k')}\right), \end{aligned} \tag{11}$$

where we have introduced the Bogoliubov dispersion $E_{c/d}(k) = \sqrt{\xi_{c/d}(k)^2 + \Delta_{d/c}(k)^2}$ and $\xi_{c/d} = \varepsilon_{c/d} - \mu$. The dependence of the interaction on the state of the bosons is encoded in the frequency structure of the interaction.

In order to estimate $T_c$ we assume a local coupling $g(q) = g$ and a flat bosonic dispersion $\Omega(q) = \Omega$. We next insert the equation for $\Delta_d$ into that of $\Delta_c$ arriving at a single self-consistent gap equation for $\Delta_c$. We find that $\Delta_c(\omega)$ exhibits only a weak frequency dependence for $\omega < \Omega$; hence, we disregard this dependence when estimating $T_c$. This allows us to evaluate all frequency sums analytically and then continue the functions to real frequencies, introducing bosonic and fermionic occupation functions. The details of the calculation and the full gap equation are given in the methods section.

Similar to the standard BCS approach for superconductivity in a single band, we only keep the most divergent terms contributing to the integrand of the gap equation. We identify two small parameters close to the Fermi surface when $|\xi_c| \ll |\xi_d| \approx \Omega$, namely the detuning between

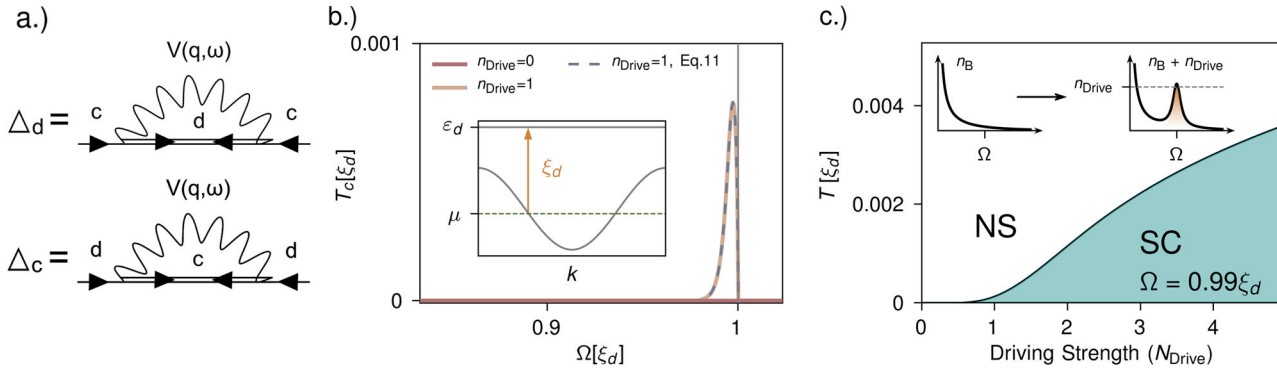

**Fig. 2 | Quantum field-theory calculations for photo-induced super-conductivity. a** Feynman diagrams for the coupled gap equations Eq. (10) and (11). **b** Critical temperature $T_c$ as a function of frequency $\Omega$ for the sawtooth chain. The dispersion of the sawtooth chain is shown in the inset with $\xi_d = \varepsilon_d - \mu$ indicated. In thermal equilibrium (red line) no significant $T_c$ is found while driving the bosons increasing their occupation by $N_{\text{Drive}}$ Eq. (16) yields a resonantly enhanced $T_c$ for $\Omega \lesssim \xi_d$ where both the full expression Eq. (26) (yellow line) and the approximate

expression Eq. (12) (blue dashed line) agree. **c** Nonequilibrium phase diagram as function of temperature $T$ and driving strength $n_{\text{Drive}}$ (Eq. (16)) in units of $\xi_d$. At a crivital driving strength the system changes from the normal state (NS) to a super conducting state (SC). The inset illustrates the non-thermal boson distribution we assume according to Eq. (16). The coupling $g$ has been set to $g = 0.2\xi_d$ throughout while the chemical potential $\mu$ is set such that $\xi_d = 3t$, where $t$ is the nearest neighbor hopping.

the upper band and the phonon frequency, $\frac{\Delta E(\xi_d)}{\Omega} = \frac{\xi_d - \Omega}{\Omega}$, and the dispersion relation of the lower band, $\frac{|\xi_c|}{\Omega}$, and expand the gap equation to leading and next-to-leading order in these parameters. In this situation $T_c$ can be determined by a gap equation of the form:

$$1 = L_{\text{BCS}}(T_c) + L_{\text{Res}}(T_c), \tag{12}$$

where the first term on the right hand side corresponds to a BCS type term,

$$L_{\text{BCS}}(T_c) = U_{BCS} \int_{-\Omega}^{\Omega} d\xi_c \, \nu(\xi_c) \frac{\tanh\left(\frac{\xi_c}{2T_c}\right)}{2|\xi_c|}, \tag{13}$$

where $\nu(\xi)$ the density of states at energy $\xi$, with an effective electron–electron attraction given by $U_{BCS} = \frac{g^4}{\Omega^3} \int_0^{2\Omega} d\xi_d \, \nu(\xi_d) (1 + n_B(\Omega) - n_F(|\xi_d|))$, which matches the attraction found from perturbation theory in the ground state (see equation (4)). Here $n_B$ is the Bose distribution function and $n_F$ is the Fermi distribution function and the integrals have a natural cut-off set by the bosonic frequency $\Omega$, since the approximate expression for the gap equation is only valid within the energy window, $|\xi_c| < \Omega$ and $|\xi_d - \Omega| < \Omega$

The second term, $L_{\text{Res}}$, is given by:

$$L_{\text{Res}}(T_c) = \frac{g^4}{\Omega^2} \int_{-\Omega}^{\Omega} d\xi_c \int_0^{2\Omega} d\xi_d \, \nu(\xi_c)\nu(\xi_d)\big(n_B(\Omega) + n_F(|\xi_d|)\big)$$
$$\frac{\Delta E(\xi_d) \tanh\left(\frac{\xi_c}{2T_c}\right) - |\xi_c| \tanh\left(\frac{\Delta E(\xi_d)}{2T_c}\right)}{|\xi_c|\left(\left(\Delta E(\xi_d)\right)^2 - |\xi_c|^2\right)}. \tag{14}$$

Intriguingly, $L_{\text{Res}}$, has no BCS analog and diverges even more strongly as $\frac{\Delta E(\xi_d)}{\Omega}, \frac{|\xi_c|}{\Omega} \to 0$. Similar to the BCS term the divergent integrand is cut-off by temperature. The appearance of a more strongly divergent term is a manifestation of the resonance found in our model when the frequency matches the energetic distance from the Fermi level to the upper band. However, at low temperatures $T \ll \Omega, |\xi_d|$, pairing from $L_{\text{Res}}$ is exponentially suppressed because it is proportional to either the number of phonons or the number of electrons in the upper band through the factor $n_B(\Omega) + n_F(\xi_d)$. This leads to the absence of a resonance effect in equilibrium consistent with the two-site model.

We now conjecture that Eqs. (12)–(14) can be used to discuss nonequilibrum systems. This is accomplished by replacing the equilibrium, thermal distribution function $n_B(\Omega)$ by a nonequilibrium

distribution function. The main limitation of this procedure is that it does not include enhanced decoherence of electrons in the presence of photoexcited bosons[30]. This decoherence provides a mechanism for pairbreaking and may result in a finite lifetime of photoinduced superconductivity[15]. A nonequilibrium boson distribution activates $L_{\text{Res}}$ and reproduces a resonantly enhanced attraction consistent with our findings in the two-site model.

## Model calculation in 1D
We use the sawtooth chain[46,47] to illustrate the interband phonon mechanism on a concrete example. This one-dimensional model consists of a dispersive lower band and a flat upper band:

$$\xi_c(k) = -\sqrt{2}t(\cos(k a) + 1) - \mu$$
$$\xi_d(k) = \sqrt{2}t - \mu. \tag{15}$$

where $a$ is the lattice constant, $t$ the hopping between sites, and $\mu$ the chemical potential. The flat upper band in the sawtooth chain gives rise to a constant detuning, providing an idealized system to identify the resonance present in our model. We explore the possibility of using the nonequilibrium boson distribution as a resource by switching on the resonant contribution in the gap equation Eq. (12) through adjusting the equilibrium boson distribution to a driven one as illustrated in Fig. 2c:

$$\langle b_q^\dagger b_q \rangle = n_B(\Omega) + n_{\text{Drive}}. \tag{16}$$

We compute $T_c$ as a function of the boson frequency $\Omega$ and explore the resonance by tuning $\Omega$ across $\xi_d$. The result is shown in Fig. 2b. In equilibrium we do not find a significant $T_c$ for the coupling of $g = 0.2\xi_d$ and the selected frequencies. However, a significant $T_c$ appears close to the resonant region as soon as we assume a non-thermal boson distribution according to Eq. (16). Both the full expression of the gap equation shown in the methods section and the approximate expression in equation (12) match precisely close to the resonance.

Unlike in the two-site model, in the extended system the driven-boson-induced interaction turns into repulsion for blue-detuned frequencies $\Omega \gtrsim \xi_d$. This is reminiscent of cavity engineered interactions between atoms that are approximately linear in the inverse cavity-pump detuning[48–50]. We note that in the blue detuned region, real transitions from the lower to the upper band, that are not captured

within our approach, become possible which are expected to be detrimental to pairing[15]. The resonant enhancement is found only for a red detuned phonon where direct electron transitions are prohibited by energy conservation, hinting that a more involved nonequilibrium treatment of the problem within the Keldysh formalism, that takes the change of Fermi functions upon driving into account, is expected to exhibit the same qualitative behavior.

In order to summarize our findings for the driven saw-tooth chain, we compute an out-of-equilibrium phase diagram as a function of temperature and driving strength (Fig. 2c), fixing $\Omega$ slightly below resonance, $\Omega = 0.99\xi_d$. While the system is in the normal state down to very low temperatures in equilibrium ($N_{Drive} = 0$) for $g = 0.2\xi_d$, populating the bosonic mode gives rise to a finite $T_c$.

### Graphene-hBN-SrTiO$_3$ heterostructure

We conclude by proposing an experimentally realizable platform for resonantly induced superconductivity, a heterostructure of graphene aligned with hBN placed on top of bulk SrTiO$_3$, as shown in Fig. 1c. Graphene aligned on hBN leads to a gap at the Dirac points of graphene, with size $\Delta E = 14$ meV[43]. The bosonic mode within our pairing model is provided by a low energy surface-phonon-polariton[44,45] that exists in the quantum paraelectric SrTiO$_3$ and couples to the interband transition in gapped graphene through its dipole moment[51,52]. We compute the relevant light-matter coupling using the mode functions obtained in ref. 53 for the surface polaritons, and approximate the resulting coupling by a box function $g(q) = \tilde{g}\,\theta(|q| - \frac{1}{d})$, where $d$ is the distance between the graphene and the SrTiO$_3$ surface which provides a natural short-wavelength cut-off for surface-phonon-polariton induced interactions. We assume an hBN thickness of 5 nm, corresponding to ~15 atomic layers[54]. We include the local Coulomb repulsion that can be estimated via density functional theory[55] and take into account Morel-Anderson renormalization processes that reduce its value to $U^* \approx 1.1$ eV which is the value we use for our estimate. Using these inputs the $T_c$ estimate is obtained by solving the full gap equation (26). The chemical potential is set such that the frequency of the surface mode is red-detuned from the inter-band transition by more than the linewidth preventing direct excitations of electrons into the upper band due to the drive.

Without any driving the Coulomb repulsion prevents pairing for reasonable temperatures. When driving the system into a state with a polaritonic occupation $n_{Drive} = 1$, we obtain $T_c = 15.2$ K—a temperature that is readily accessible. Assuming that only polariton modes with momentum $q < \frac{1}{d}$ are excited, the required polaritonic occupation corresponds to exciting $3 \times 10^{-4}$ bosons per graphene unit cell.

A distinctive test of our theory may be performed by further reducing the chemical potential in graphene which, according to the proposed mechanism, is expected to decrease $T_c$ despite resulting in a larger density of states at the Fermi-surface.

In the Methods section, we estimate the change in temperature due to heating that the sample would undergo if all of the energy of the initial excitation was converted into heat close to the surface of the SrTiO$_3$ substrate. We find that at the critical temperature $T_c = 15.2$ K the expected temperature change would be less than 1 K. This emphasizes that the driving we propose is of realistic strength. Surface modes cannot be directly excited by a laser due to their dispersion lying outside the light-cone. However, these modes can be driven by irradiating a tip, which generates excitations in the near-field that effectively drive the modes[56].

A final comment is in order regarding the assumed pairing symmetry of resonantly induced superconductivity. In this work we considered only s-wave superconductivity. In reality, due to the electronic Coulomb repulsion, d-wave pairing might be favored, and our mechanism can equally well lead to or contribute to d-wave superconductivity. In such a scenario, the negative effect of the local Coulomb repulsion would be further mitigated.

## Discussion

We introduced a simple microscopic mechanism based on a boson coupled to an electronic interband transition. We demonstrated both photo-induced enhancement of superconductivity, as well as resonant amplification of this enhancement when the phonon frequency is red-detuned but nearly aligned with the electronic excitation energy between the Fermi level in the partially filled lower band and the empty upper band. We demonstrated the existence of the resonantly amplified electron attraction by both exact diagonalization for a two-site model and by deriving a gap equation for extended systems. Tantalizingly, we showed that this interaction can be engineered in a two-dimensional material on a surface-phonon-polariton platform. For an example heterostructure of graphene-hBN-SrTiO$_3$, we computed that superconductivity can arise at temperatures around $T_c \approx 15$ K. For an example heterostructure of graphene-hBN-SrTiO$_3$ that superconductivity can arise at temperatures around $T_c \approx 15$ K upon driving. Owing to the microscopic nature of our mechanism, we are able to make a prediction without relying on free parameters.

We believe that our simplified treatment of the nonequilibrium problem correctly captures the renormalization of the effective electron-electron interaction due to a non-thermal distribution of bosons. However, we do not exclude that heating and pair-breaking effects counteracting superconductivity[15,17] might be underestimated here, in particular close to the resonance, which might create a trade-off that needs to be taken into account when tuning the frequency. Moreover, dynamical generation of disorder that was theoretically shown to limit the possibility to realize light induced super-conducting like states in 1D systems, was not explored here either[57]. It is worth noting, however, that in 2D systems, several studies have argued that disorder leads to multi-fractality that actually increases the transition temperature of superconductivity[58]. The interplay between disorder and resonantly photo-induced superconductivity is thus left for future research.

Interestingly, even though the enhancement of the effective electron-electron attraction is likely a transient phenomenon, the superconducting state arising from it might be longer lived than the initial drive, as indicated by the accompanying study of the authors on the same model away from the resonance[59].

In view of the resonant nature of our mechanism, an interesting future direction is to explore its applicability in the context of cavity materials engineering[60–64], for which several proposals for cavity-induced superconductivity already exist[65–69].

## Methods

### Schrieffer-Wolf transformation

We perform a Schrieffer-Wolf transformation of the local Hamiltonian Eq. (3) according to

$$H^{SW} = e^S H^{loc} e^{-S} \tag{17}$$

using as transformation matrix[70]

$$\begin{aligned}
S = &\frac{\alpha}{\Omega}\sum_j \left(b_j^\dagger - b_j\right)\sum_\sigma \left(d_{j,\sigma}^\dagger c_{j,\sigma} + c_{j,\sigma}^\dagger d_{j,\sigma}\right) \\
&+ i\frac{\beta}{\Omega}\sum_j \left(b_j^\dagger + b_j\right)\sum_\sigma \left(d_{j,\sigma}^\dagger c_{j,\sigma} - c_{j,\sigma}^\dagger d_{j,\sigma}\right).
\end{aligned} \tag{18}$$

with $\alpha$ and $\beta$ to be determined by the condition

$$[S, H_0] = -H_{int}. \tag{19}$$

This condition is satisfied when

$$
\begin{aligned}
\alpha &= \frac{g\Omega^2}{\Delta E^2 - \Omega^2} \\
\beta &= -i\frac{g\Omega\Delta E}{\Delta E^2 - \Omega^2}
\end{aligned}
\tag{20}
$$

The effective Schrieffer-Wolff Hamiltonian to order $g^2$ are calculated via $\frac{1}{2}[S,V]$ which gives rise to Eq. (5) in the results section.

## Attractive correlations in 2-site model

In order to explore the resonant regime $\Omega \approx \Delta E$ for the model in Eq. (7) we perform full diagonalization calculations. We limit the bosonic part of the Hilbert space to a maximumm of $N_{max} = 6$ bosons for which all quantities converge. To explore the out of equilibrium properties, we add a driving term to the Hamiltonian according to Eq. (9). In Eq. (9), $F(t)$ has a normalized Gaussian shape centered around $t_{pump} = \frac{16\pi}{\omega_D}$ and standard deviation $s = \frac{4\pi}{\omega_D}$ multiplied by a bare driving strength $F_0$ that is set to $F_0 = \frac{\Delta E}{2}$. We always drive the bosons on resonance setting $\omega_D = \Omega$. We evolve the system in time with the time evolution operator $U(t) = \mathcal{T}\exp(i\int_{t_0}^{t_1}H(t)dt)$, where $\mathcal{T}$ is the time-ordering operator, using finite time steps of width $\Delta t = \frac{\pi}{10\omega_D}$, i.e., 20 time steps per driving period. We compute the time-averaged local doublon correlations

$$
\bar{C} = \int_{t_0}^{t_1}\langle n_{j,\uparrow}^c n_{j,\downarrow}^c\rangle(t) - \langle n_{j,\uparrow}^c\rangle(t)\langle n_{j,\downarrow}^c\rangle(t)\, dt,
\tag{21}
$$

where $t_0 = \frac{30\pi}{\omega_D}$ and $t_1 = \frac{130\pi}{\omega_D}$, reported in Fig. 1b together with the time-averaged boson number. We show a concrete time-evolution for longer times in Fig. 3 for the resonant case $\Omega = \Delta E$ and a slightly red-detuned case $\omega_D = 0.9\Delta E$. One can see that all quantities undergo heavy oscillations that are not decaying attributed to quantum revivals for this very small system that we do not expect to be present in extended systems. However, the time-averaged correlations quickly converge to a finite value. We additionally mark the time interval over which we computed the time average in the main part $[t_0, t_1]$. We average over 50 driving periods after the pump has decayed. The reason we limit the averaging time is that in a real system with dissipation the excitation would decay—presumably not reaching 50 oscillation cycles. Reducing the number of cycles for the averaging process does not qualitatively change our results.

## Effective electron-electron interaction

We use the path-integral formalism to derive a dynamic electron-electron interaction from the model Eq. (2). The bare action of the

system, including bosonic modes, can be written as

$$
S_0 = \sum_q \bar{b}_q D(q)^{-1} b_q + \sum_{k,\sigma,n} \bar{\psi}_{k,\sigma,n} G_0^{-1}(k,n)\psi_{k,\sigma,n}
\tag{22}
$$

where $k = (\vec{k}, i\omega)$ and $q = (\vec{q}, i\Omega)$ are composite indices of a lattice momentum $\vec{k}$ or $\vec{q}$ (we distinguish the composite index from the lattice momentum by explicitly denoting the lattice momentum by a vector) and a fermionic(bosonic) Matsubara frequency $\omega(\Omega)$. The $b_q$ are complex field, $\bar{b}_q$ their complex conjugate and $D(q)^{-1} = i\Omega - \Omega(\vec{q})$ is the inverse bosonic propagator. $\psi_{k,\sigma,n}$ and $\bar{\psi}_{k,\sigma,n}$ are Grassmann valued fields with spin index $\sigma \in \{\uparrow\downarrow\}$ and band-index $n \in \{c, d\}$ and $G_0^{-1}(k,n) = i\omega - \varepsilon_n(\vec{k})$ is the inverse bare electronic Green's function. We introduce an inter-band coupling via the bosonic fields as

$$
S_{int} = \sum_q g(\vec{q})\left(b_q + \bar{b}_{-q}\right)\left(\sum_{k,\sigma,n}\bar{\psi}_{k+q,\sigma,n}\psi_{k,\sigma,\bar{n}}\right).
\tag{23}
$$

Assuming a 2-band model we have introduced $\bar{n}$ as the band complementing that labeled by the band index $n$. $g(\vec{q})$ is the $\vec{q}$ dependent inter-band coupling. Now we can integrate out the bosons with a suitable substitution of the bosonic fields in the path integral to arrive at an effevtive interaction

$$
S_{int}^{eff} = -\sum_q g(\vec{q})g(-\vec{q})D(q) \\
\left(\sum_{k,\sigma,n}\bar{\psi}_{k+q,\sigma,n}\psi_{k,\sigma,\bar{n}}\right)\left(\sum_{k',\sigma',m}\bar{\psi}_{k'-q,\sigma',m}\psi_{k',\sigma',\bar{m}}\right).
\tag{24}
$$

Due to the sum over $q$ and since all other terms are even in $q$ one may perform the replacement

$$
D(q) \to \frac{-\Omega(\vec{q})}{\Omega^2 + \Omega(\vec{q})^2}
\tag{25}
$$

taking only the part of the propagator even in $q$. Eqs. (10) and (11) are derived using the saddle point approximation in the Cooper channel, leading to a self-consistent equation shown schematically in Fig. 2a. A key aspect of our work is performing the Frequency sums analytically and expressing the result in terms of occupation distributions of the bosons and fermions. In the main text we show an approximate formula of the most divergent terms, for completeness we report here the full gap equation. We take the full $\omega$ and $k$ dependence of $\Delta_d$ by replacing the expression for $\Delta_d$ in Eq. (11) into Eq. (10), to arrive at a self-consistent equation for $\Delta_c$ only. We then find that the expression

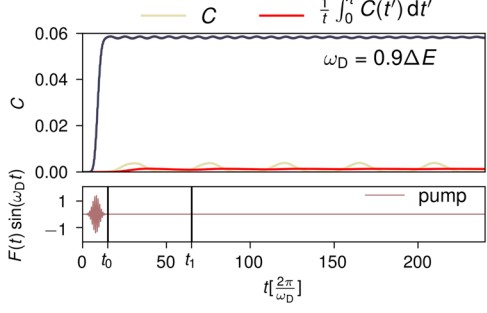
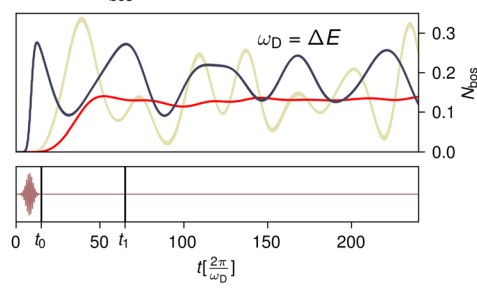

**Fig. 3 | Real-time evolution of the two-site model.** Time evolution of $N_{bos}$ (blue line, right axis) and $C$ (yellow line, left axis) red detuned from the resonance $\Omega = 0.9\Delta E$ and at resonance $\Omega = \Delta E$ for long times of the system defined by Eq. (9). The times $t_0$ and $t_1$ used for the time average in Eq. (21) are marked in the bottom panel where the pump envelope according to Eq. (9) is shown. Additionally we show the time averaged correlations as a function of the upper time of the averaging. This quantity, which underlies Fig. 1b) in the main part, quickly converges.

for $\Delta_c$ is weakly frequency dependent for $\omega \ll \Omega(\vec{q})$ which allows to approximate $\Delta_c$ as indpendent of $\omega$ and $\vec{k}$. The resulting gap equation that determines $T_c$ is found to be:

$$
1 = \frac{1}{N^2} \sum_{\vec{k}',\vec{k}} |g|^4 \frac{1}{E_1 E_2} \times \left( \frac{(2n_f(E_1)-1)(E_1^2(2E_2 n_b(\Omega)+E_2-2n_f(E_2)\Omega+\Omega)}{\Omega(\Omega^2-E_1^2)((E_2-\Omega)^2-E_1^2)((E_2+\Omega)^2-E_1^2)} + \right.
$$

$$
\frac{(2n_f(E_1)-1)((\Omega-E_2)(E_2+\Omega)(2E_2 n_b(\Omega)+E_2+(2n_f(E_2)-1)\Omega))}{\Omega(\Omega^2-E_1^2)((E_2-\Omega)^2-E_1^2)((E_2+\Omega)^2-E_1^2)} \bigg)
$$

$$
+ \frac{4E_1 E_2 (n_f(\Omega+E_2)-1)(E_2+\Omega)(n_b(\Omega)-n_f(E_2)+1)}{-4E_2^2 \Omega(E_2+\Omega)(E_2+2\Omega)((E_2+\Omega)^2-E_1^2)}
$$

$$
+ \frac{4E_1 E_2 (n_f(\Omega-E_2)-1)(E_2-\Omega)(n_b(\Omega)+n_f(E_2))}{4E_2^2 \Omega(E_2-2\Omega)(E_2-\Omega)((E_2-\Omega)^2-E_1^2)}
$$

$$
+ \frac{-4E_1 E_2 n_f(\Omega-E_2)(E_2-\Omega)(n_b(\Omega)+n_f(E_2))}{-4E_2^2 \Omega(E_2-2\Omega)(E_2-\Omega)((E_2-\Omega)^2-E_1^2)}
$$

$$
+ \frac{4E_1 E_2 n_f(\Omega+E_2)(E_2+\Omega)(n_b(\Omega)-n_f(E_2)+1)}{-4E_2^2 \Omega(E_2+\Omega)(E_2+2\Omega)((E_2+\Omega)^2-E_1^2)}
$$

$$
+ \frac{E_1 E_2 (n_b(\Omega)+1)(E_2^2(2n_b(\Omega)+1)+E_2(2n_f(E_2)-1)\Omega-2(2n_b(\Omega)+1)\Omega^2)}{\Omega^2(E_1^2-\Omega^2)(E_2^4-4E_2^2 \Omega^2)}
$$

$$
\left. \frac{E_1 E_2 n_b(\Omega)(E_2^2(2n_b(\Omega)+1)+E_2(2n_f(E_2)-1)\Omega-2(2n_b(\Omega)+1)\Omega^2)}{\Omega^2(E_1^2-\Omega^2)(E_2^4-4E_2^2 \Omega^2)} \right).
$$

(26)

Here $E_1 = E_1(\vec{k})$ and $E_2 = E_2(\vec{k}')$ denote the Bogoliubov quasi-particle dispersion in the two considered bands where we have omitted the quasi-momentum dependence in Eq. (26) for brevity.

### Graphene coupled to surface phonon polaritons

In this part we outline the estimate of the pairing temperature of our proposed heterostructure consisting of Graphene aligned with hBN on top of bulk $SrTiO_3$. We use the $q$-dependent surface phonon-polariton modes given in ref. [53] that read

$$
\phi(z>0) = N \left( \frac{q_x}{|q_{xy}|}, \frac{q_y}{|q_{xy}|}, i \sqrt{\left| \frac{\varepsilon(\omega)}{\varepsilon_{HBN}} \right|} \right)^T
$$

$$
e^{-\sqrt{\left| \frac{\varepsilon_{HBN}}{\varepsilon(\omega)} \right|} |q_{xy}|z} e^{i(q_x x + q_y y - \omega t)}
$$

$$
\phi(z<0) = N \left( \frac{q_x}{|q_{xy}|}, \frac{q_y}{|q_{xy}|}, -i \sqrt{\left| \frac{\varepsilon_{HBN}}{\varepsilon(\omega)} \right|} \right)^T
$$

$$
e^{\sqrt{\left| \frac{\varepsilon(\omega)}{\varepsilon_{HBN}} \right|} |q_{xy}|z} e^{i(q_x x + q_y y - \omega t)}
$$

(27)

where the upper half-space $z > 0$ is assumed to be filled with hBN with dielectric constant $\varepsilon_{hBN} = 5$[71] while the lower half-space is made up of the $SrTiO_3$ with $\varepsilon(\omega) = \varepsilon_\infty \frac{\omega_{LO}^2 - \omega^2}{\omega_{TO}^2 - \omega^2}$. We use $\varepsilon_\infty = 6.3$[72], $\omega_{TO} = 2\pi \times 1.26\,THz$ and $\omega_{LO} = 2\pi \times 5.1\,THz$[73]. The normalization $N$ of the modes can be determined from the normalization condition[53]

$$
\hbar \Omega_s \varepsilon_0 \int_V dr\, \varepsilon(\Omega_s) \nu(\Omega_s) |\phi(r)|^2 = 1
$$

(28)

with $\nu(\omega) = 2 + \omega \partial_\omega \varepsilon(\omega)$ and $\varepsilon_0$ the vacuum permittivity. The limiting surface phonon frequency when $c^2 q^2 \gg \omega_{LO}^2$ is $\Omega_s = 2\pi \times 3.9\,THz$ which we will use throughout for our estimate. In order to estimate the light-matter coupling we focus on the Dirac cone at which the Hamiltonian, including the gap opening due to an effective onsite potential induced through the proximity of hBN to the graphene layer, reads

$$
H = \frac{V_0}{2} \sigma_z + \hbar \nu_F (k + eA) \cdot \sigma.
$$

(29)

where $V_0$ is the onsite potential, $\nu_F$ the Fermi velocity and $A$ the vector potential due to the surface polaritons. At the $K$-point the inter-band coupling is equal to $g_{inter} = 1$ while far away from the $K$-point it can be estimated from the Dirac Hamiltonian without including an onsite potential which yields $g_{inter} = \frac{i \hbar \nu_F (k \times A)}{|k|}$. Using the coupling away from

the $K$-point as a conservative estimate, we get for the inter-band coupling

$$
g^{k,q} = i \frac{\sqrt{4\pi\alpha}}{\sqrt{N}} f(\omega) \frac{\hbar \nu_F \sqrt{c}}{\sqrt{\Omega_s S_{cell}}} \frac{(k \times q)}{|q||k|} \sqrt{|q_{xy}|} e^{-|q_{xy}|z}
$$

(30)

where $f(\omega) = \left( 4 \frac{\omega^2(\varepsilon_\infty + \varepsilon_{HBN})}{\omega^2 - \omega_{TO}^2} \right)^{-\frac{1}{2}}$, $S_{cell}$ is the area of the unit cell in graphene, $\alpha$ the fine-structure constant and $c$ the speed of light. In order to obtain a simple estimate we approximate the coupling by a box function $g^{k,q} \to \tilde{g}\theta(|q| - \frac{1}{d})$. The cutoff $\frac{1}{d}$ is naturally provided by the mode-function Eq. (27) since $\varepsilon(\omega) \to -\varepsilon_{hBN}$ for $q^2 c^2 \gg \omega_{LO}^2$ and therefore $|\phi(q)|^2 \sim e^{-2qd}$. We determine $\tilde{g}$ by fixing the integral

$$
\tilde{g}^2 = \int_{\mathbb{R}^2} |g^{k,q}|^2 \, dq = \frac{2\pi\alpha}{3\sqrt{3}N} f(\omega)^2 \frac{\hbar^2 \nu_F^2 c}{a^2 d}
$$

(31)

where we use $g^2$ since this appears in the interaction. Due to the box-function the coupling has now attained a $q$ dependence in principle leading to a $k$-dependence of the gap itself. We compute the gap at $k = K$ and assume it to be zero outside of the coupling radius $\theta(k)$. Additionally, the momentum transfer $k - k'$ is limited due to the box coupling by a factor $\theta(k - k')$. We set the chemical potential $\mu = -15\,meV$ slightly into the lower band and otherwise the parameters from the main text. This choice of chemical potential leads to a finite carrier density in the lower band and corresponds to an interband optical transition threshold of 22 meV, preventing any direct dipole transitions due to the 16.1 meV surface phonon polariton oscillations. To prevent heating from direct dipole transitions, the surface phonon polariton must be red-detuned from the optical transition threshold by more than the phonon's linewidth, which in $SrTiO_3$ is around 10%[74]. We include the local Coulomb repulsion which was estimated to be $U \approx 9.3\,eV$ based on density functional theory calculations[55]. The bare local repulsion will be reduced by Morel-Anderson renormalization according to[75]

$$
U^* = \frac{U}{1 + \frac{2U}{W} \ln\left(\frac{W}{2\Omega_s}\right)} \approx \frac{U}{8.1} \approx 1.1\,eV,
$$

(32)

which we take as the value of the effective Hubbard $U$ for our $T_c$ estimate. Importantly, the resulting $T_c$ does not depend very sensitively on the precise value of $U^*$. Due to the exponential dependence of the polariton mode field strength on the distance $d$ from the surface, $T_c$ does, however, vary significantly with $d$. In particular, short distances are very desirable to achieve a higher $T_c$. We evaluate Eq. (26) on a $k$-grid of size $500 \times 500$. For the driven case we assume a non-thermal boson occupation according to Eq. (16), setting $n_{Drive} = 1$ for all modes with $q < \frac{1}{d}$.

### Estimated heating due to the drive

In order to quantify the strength of the drive we propose to induce superconductivity in the graphene-HBN-STO heterostructure, we provide an order of magnitude estimate of the heating of the sample that would result from the drive. The surface modes of STO couple to bulk modes through phonon non-linearities and disorder and therefore heat up the substrate. For a first estimate, we assume this to be the dominant heating effect and hence estimate the heating of the STO substrate assuming all energy of the initial excitation to be converted into heat. Shortly after the drive, the surface mode will only heat up a thin layer of the substrate close to the surface corresponding to the penetration depth which sets the length scale over which the surface modes couple to the bulk. At later times the heat will dissipate. We take $\xi = 5\,nm$ as the penetration depth corresponding to the penetration depth of the mode with the largest $q$ of all modes that we assume to have a non-zero coupling to graphene (see previous section). This constitutes a conservative estimate since modes with smaller $q$ have a

larger penetration depth. To get an estimate of the energy per volume, we compute expected number of excitations within the area of one unit-cell of graphene

$$n_{\text{ex}} = \frac{\int_{\text{BZ}} b_q^\dagger b_q \, \mathrm{d}q}{\int_{\text{BZ}} 1 \, \mathrm{d}q} = \frac{\int_{\text{BZ}} n_{\text{Drive}} \Theta(\frac{1}{d} - q) \, \mathrm{d}q}{\int_{\text{BZ}} 1 \, \mathrm{d}q} \approx 3 \times 10^{-4} \tag{33}$$

where in the second step we have assumed that only modes with small wave-vectors are excited for which we chose the same cutoff as for the momentum coupling and integrals run over the Brillouin zone. The expected change in temperature is then

$$\Delta T = \hbar \Omega_s \frac{n_{\text{ex}} V_{\text{STO}}}{\xi A_C} \frac{N_{\text{av}}}{c_P} \tag{34}$$

where $V_{\text{STO}}$ is the unit-cell volume of STO, $A_C$ the area of the unit-cell of graphene, $N_{\text{av}}$ the Avogadro number and $c_P$ the specific heat of STO at constant pressure. We take the specific heat of STO at 15K from ref. 76 as $c_p/T^3 \approx 0.75 \times 10^{-4} \frac{\text{J}}{\text{mol K}^4}$ and use as the lattice constant $a_{\text{STO}} = 0.3905$ nm. Since STO has cubic lattice structure we have $V_{\text{STO}} = a_{\text{STO}}^3$. Put together we obtain a change in temperature of

$$\Delta T = 0.7 K. \tag{35}$$

On the other hand, for an estimate of an upper bound on the induced heating, one may assume that all energy of the initial drive is converted into heat solely in the graphene flake. For this estimate we use the heat capacity of graphene $c_P = 0.24 \frac{\text{J}}{\text{mol K}}$ at $T = 10$ K taken from ref. 77 and compute

$$\Delta T = \frac{\hbar \Omega_s n_{\text{ex}}}{c_P} \approx 1.9 K. \tag{36}$$

This is likely an overestimate, but given that we assume a temperature of the sample of 10 K and computed $T_c \approx 15$ K, the proposed drive would not destroy the superconductivity solely by heating the sample.

## Data availability
The data presented in this work is produced directly by the codes provided below. The data can also be provided upon request.

## Code availability
The codes used to produce the data in the paper are openly available on github including instructions for installation and usage. The exact diagonalization code can be found under ref. 78, the Mathematica notebook solving the coupled gap equations in, Eqs. (10) and (11) arriving at Eq. (26) under ref. 79 and the numerical code solving the gap equations under ref. 80.

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

## Acknowledgements
We acknowledge fruitful discussion with Jonathan B. Curtis, Mohammad Hafezi, Andrey Grankin, Daniele Guerci, Angel Rubio, John Sous, Andy Millis, Martin Eckstein and Hope Bretscher. M.H.M. is grateful for the financial support received from the Alex von Humboldt postdoctoral fellowship. S.C. is grateful for support from the NSF under Grant No. DGE-1845298 & for the hospitality of the Max Planck Institute for the Structure and Dynamics of Matter. E.A.D. acknowledges support from the ARO grant "Control of Many-Body States Using Strong Coherent Light-Matter Coupling in Terahertz Cavities' and the SNSF project 200021_212899.

## Author contributions
M.H.M. conceived the project together with C.J.E., M.A.S. and E.A.D. C.J.E. and M.H.M. developed the theoretical and analytical framework and analyzed data. C.J.E. developed and performed all numerical simulations. M.H.M., M.A.S. and D.M.K. supervised the project. All authors participated in the discussion and interpretation of the results. C.J.E., M.H.M., M.A.S. and S.C. wrote the manuscript with input from all authors.

## Funding

## Competing interests
The authors declare no competing interests.
