## [Peer Review File · Nature Communications]

Theory of resonantly enhanced photo-induced superconductivityREVIEWER COMMENTS

Reviewer #1 (Remarks to the Author):

The authors present a theoretical description of a model that may be relevant to photo-induced superconductivity. In particular, they consider a previously proposed model where a boson is coupled to an electronic interband transition, rather than an electronic population as in standard electron-phonon coupling models. Such strong coupling has been analyzed before in conjunction with time-resolved experiments that study non-equilibrium superconductivity. If realized in a physical system, it will modify the electron-electron interaction under certain regimes. This process will be enhanced if the underlying boson is driven into a suitable non-thermal state and when its frequency is tuned close to the interband transition. Such models have been proposed before as one possible mechanism to explain the controversial and much debated but highly interesting topic of light-induced superconductivity, and may be relevant to explain certain experimental features observed before.

In that sense, a theoretical paper like this one studying a model Hamiltonian are suitable for publication in a technical journal. This is even more the case given the debate and efforts invested in the experimental realization of light-driven superconductivity. The much needed guidance of experiments requires a detailed analysis of realistic models and their direct relation to the experimental measurements. At this stage of the field, it is important to predict smoking gun experimental features that would validate the proposed models. This requires solving the proposed models under conditions that are directly related to the experimental conditions. While experiments have been motivated by simple models like the present one in the past, so far there is no conclusion, perhaps due to the lack of specific guidance for specific experiments. While the proposal of a graphene-hBN-SrTiO₃ heterostructure and the estimate of a superconducting T_c that may be achieved upon driving the system are interesting, I believe that, at this stage of the field and given past literature and experimental attempts, this paper and level of theory used are more suitable for publication in a technical journal.

In summary, it is my opinion that a closer and more realistic connection between theory and ultrafast spectroscopy experiments is needed to make progress at this stage of the field, given past reports and controversies. While such theory-experiment effort is certainly appropriate for Nature journals, it can be guided by simple models like the present one analyzed in more technical publications.

Reviewer #2 (Remarks to the Author):

In this manuscript, Eckhardt et al. proposed a mechanism of light enhanced superconductivity. A phonon that induces an electronic interband transition mediates an intraband attraction between the electrons. Since this process is proportional to the phonon occupation number, the authors propose that photo excitation of the phonon enhances this attraction. This effect is also resonantly enhanced when the phonon frequency approaches the interband threshold. The authors applied it to the sawtooth chain model and to a realistic graphene-SrTiO₃ heterostructure. This topic is of substantial interest. I would recommend its publication given that the following questions and issues are addressed:

1, If one goes directly from the second term in Eq. 5 to Eq. 6, the occupation factor in Eq. 6 should be $b^{b+1/2}$ instead of b^b , similar to Ref. 14. This means that the resonance behavior should occur in equilibrium too. This is not the case found by the authors, as shown in Fig. 1b and discussed in the text. Why?

2, In the estimates for the graphene-SrTiO₃ device, how large is the interband optical transition threshold. Is it above or below the surface phonon frequency (16.1 meV) of SrTiO₃? Above Eq. 32, it is said that the chemical potential is -15 meV, does it mean that the interband threshold is 30 meV?

3, The SrTiO₃ phonon has a large linewidth, giving a large energy uncertainty. This probably means that the effective detuning between the phonon frequency and the interband threshold can never be very small. How does this fact change the estimations?

4, If the SrTiO₃ surface phonon frequency is above the interband threshold of graphene, the electrons in graphene are heated by the pump too. This may cause a serious heating given the much lower heat capacity of electrons. Is this an issue?

5, Typo in Fig. 2b: "Eq. 11" should probably be "Eq. 12".

Response to Reviewer comments and list of changes at the end :

REVIEWER COMMENTS

Reviewer #1 (Remarks to the Author):

The authors present a theoretical description of a model that may be relevant to photo-induced superconductivity. In particular, they consider a previously proposed model where a boson is coupled to an electronic interband transition, rather than an electronic population as in standard electron-phonon coupling models. Such strong coupling has been analyzed before in conjunction with time-resolved experiments that study non-equilibrium superconductivity.

Our response: We thank the referee for their time and effort reviewing the paper and giving us the opportunity to clarify that the mechanism we propose for light-induced superconductivity is indeed a **new mechanism**. While it is true that there are few works where this model has been considered in the context of equilibrium superconductivity in SrTiO₃ (Ref. 34-37 of our manuscript), its non-equilibrium properties are, to the best of the authors' knowledge, unexplored in the context of light-induced superconductivity. As a matter of fact, none of the previous theory works on light-induced superconductivity cited in our manuscript (Refs. 14-33) discuss the model of our paper (If the reviewer has a reference in mind that considers the out of equilibrium dynamics of this model we would gladly consider it).

We try to make this point more clear by adjusting the beginning of the second paragraph of the introduction that now reads

“In the context of equilibrium superconductivity in doped SrTiO₃,^{34–37} the importance of local interband- phonon coupling has previously been discussed. In this paper, however, we highlight the non-equilibrium properties of this model that to the best of the authors' knowledge have not been explored to date in the context of driven superconductivity. This can potentially elucidate the microscopic mechanism behind photoinduced superconductivity”

We hope that this adjustment clarifies the novelty of our results.

If realized in a physical system, it will modify the electron-electron interaction under certain regimes. This process will be enhanced if the underlying boson is driven into a suitable non-thermal state and when its frequency is tuned close to the interband transition. Such models have been proposed before as one possible mechanism to explain the controversial and much debated but highly interesting topic of light-induced superconductivity, and may be relevant to explain certain experimental features observed before. In that sense, a theoretical paper like this one studying a model Hamiltonian are suitable for publication in a technical journal. This is even more the case given the debate and efforts invested in the experimental realization of light-driven superconductivity. The much needed guidance of experiments requires

a detailed analysis of realistic models and their direct relation to the experimental measurements.

Our response:

The reviewer provides an interesting discussion and viewpoint of this challenging field. Indeed, we fully agree with the Reviewer that in order to make progress in this field, the theory needs to be grounded in real systems and make concrete predictions that can be tested experimentally. In fact, this concern has been at the center of our theoretical investigation and we believe that in this regard it stands out compared to other theory works. We point to this issue in the original manuscript in the intro:

“A considerable amount of theoretical effort has been directed at understanding photo-driven states, with a variety of proposals attempting to explain the phenomenology of photo-induced superconductivity. However, a simple microscopic and experimentally realistic mechanism that predicts photo-controlled superconductivity, able to direct future experimental explorations, remains elusive.”

We believe that the connection of our mechanism to experiments and real systems is one of the main strengths of our work that distinguishes it from previous theoretical explorations. First, we will explain the measures we have taken to make sure that our theory is experimentally relevant and has a high chance of being observed, second we will contrast our theory with previous theoretical proposals on photo-induced superconductivity focusing on realism and experimental relevance, and finally we will explain our strategy for pushing the field of light-induced superconductivity forward using 2D heterostructures:

- 1) To make sure that our results will be experimentally relevant:
 - a) We have first solved the out-of-equilibrium dynamics using a range of different theoretical and numerical methods to confirm that the resonant enhancement of attraction is real and not an artifact of our theoretical approach.
 - b) Then, we have developed the theory of graphene coupled to SrTiO₃ surface phonon polaritons. This was done in order to provide realistic estimates of photo-induced superconductivity with this mechanism using widely accessible material components **with no free parameters**. All the inputs for the theory were either data from first principles calculations or experiments.
 - c) Usually, theoretical proposals are hampered by heating which competes with superconductivity during driving. In the Methods section we have checked and confirmed that the heating effects on graphene and SrTiO₃ for the driving we propose will be very small. In fact, a major selling point for our mechanism is that the resonant enhancement allows photo-induced superconductivity for rather gentle driving, making it even more likely to appear in the experimental set-up we propose.
- 2) We would like to re-emphasize that our proposed mechanism is new and unexplored. Moreover, unlike previous work on light-induced superconductivity quoted in our

introduction which is mostly phenomenological, our mechanism is truly microscopic, simple and ubiquitous. For example in the case of SrTiO₃ coupled to graphene, the interband matrix element is the dipole transition matrix element which can be computed from first principles. This is in stark contrast with other works in the theory for light induced superconductivity that typically don't make any estimates at all, and do not make concrete experimental proposals. This is either because the model is not sufficiently microscopic or because the proposed interaction is hard to realize, unlike our work.

- 3) Explaining current light-induced superconductivity experiments is very hard. One of the primary reasons for this is that the materials involved are intractable and strongly correlated even in equilibrium: High T_c cuprates, K₃C₆₀, and kappa-salts. Instead, we propose a new strategy for this field, using 2D heterostructures to build simple controllable set-ups that implement a concrete microscopic theory proposal. The possibility to create new photo-induced superconducting examples using 2D materials placed on top of surfaces is a novel idea which forms an integral part of our paper. We clarify this point further by changing the abstract to read:

“We discuss two-dimensional heterostructures as a potential test ground for light-induced superconductivity concretely proposing a setup consisting of a graphene-hBN-SrTiO₃ heterostructure, for which we estimate a superconducting T_c that may be achieved upon driving the system.”

We have also expanded on this point in the introduction, modifying the first sentences of the last paragraph that now read

“Finally we put forward two-dimensional heterostructures as a test-bed for our theory. In particular, we propose to couple two-dimensional materials to the surface phonon polaritons of a substrate. Previous materials, in which light-induced superconductivity has been observed like cuprates,³⁸ κ-salts¹³ and K₃C₆₀,^{10–12} are strongly correlated hindering the formulation of simple theories as well as providing limited experimental control handles, making the verification of proposed mechanisms difficult. In contrast, two-dimensional (van der Waals) materials as well as surface phonon polaritons offer precise control potentially enabling direct tests of concrete microscopic theory proposals for light-induced superconductivity.”

We feel that the concrete experimental set-up that our manuscript proposes for light-induced superconductivity in general and our specific theory in particular are more clearly emphasized in the updated version of the paper – we therefore thank the referee for stressing this important point.

At this stage of the field, it is important to predict smoking gun experimental features that would validate the proposed models. This requires solving the proposed models under conditions that are directly related to the experimental conditions.

Our response:

We agree with the referee that concrete experimental predictions are needed to validate theory proposals for light-induced superconductivity. We would like to emphasize that in our view our work constitutes a significant advancement in this direction.

As stated before, rather than post-dicting existing experiments, or providing a phenomenological mechanism that cannot easily be used to make predictions, the microscopic mechanism we provide allows us to theoretically investigate and make predictions about **light-induced superconductivity in 2D heterostructures**. In particular, the stack we propose of SrTiO₃ - hBN - graphene with a tip exciting surface plasmons is very realistic with current experimental technologies and very specific.

We agree with the referee that conclusively proving the presence of photo-induced superconductivity in experiments is an important step that needs to be addressed by theory and experiments. This daunting task is being pursued with new ideas in nonlinear optical spectroscopy and time-resolved magnetometry. However, that is beyond the scope of our manuscript as it would greatly deviate from the main message of the paper focusing on the mechanism rather than detection. However, given the Reviewer's comments we have provided an additional distinctive feature that may be considered a "smoking gun" for our mechanism : We propose to investigate the dependence of the light-induced state on the position of the chemical potential. Further lowering the chemical potential, i.e. further red-detuning it from band transitions, is expected to lower the critical temperature according to our mechanism. This is indeed a non-trivial prediction, as the density of states at the Fermi-surface increases when lowering the chemical potential. We have included this prediction into the Results part of the updated version of the manuscript where we now write

"A distinctive test of our theory may be performed by further reducing the chemical potential in graphene which, according to the proposed mechanism, is expected to decrease T_c despite resulting in a larger density of states at the Fermi-surface."

We would like to emphasize that such experimental control and thus verification of this theory prediction is not available in current materials that show features of light-induced superconductivity highlighting the advantages of our proposal to investigate light-induced superconductivity in 2D heterostructures. As a last comment we would like to point out that we have indeed solved the model under experimental conditions: No free parameters enter the calculation; all parameters are taken from ab-initio simulations or experiments.

The addition of the above arguments has clearly strengthened our paper, illustrating the advantages of considering 2D heterostructures plus suggesting finger-print features of the mechanism we propose. We would therefore like to thank the referee for bringing up this point.

While experiments have been motivated by simple models like the present one in the past, so far there is no conclusion, perhaps due to the lack of specific guidance for specific experiments. While the proposal of a graphene-hBN-SrTiO₃ heterostructure and the estimate of a superconducting T_c that may be achieved upon driving the system are interesting, I believe that, at this stage of the field and given past literature and experimental attempts, this paper and level of theory used are more suitable for publication in a technical journal.

Our response: We would like to thank the referee for judging our experimental proposal as interesting. We agree with the referee that specific guidance to experiment is dearly needed in the field at this point. We believe that our paper makes a valuable contribution in this direction in two ways: Firstly, it provides a simple and microscopic mechanism for light-induced superconductivity that was previously not available. This enables genuine experimental predictions without any free parameters – which we have provided (note that the driving strength of the proposed experiment is fixed via the heating estimate i.e. the total energy of the drive).

To clarify this point we have added a sentence to the discussion section reading

“Owing to the microscopic nature of our mechanism, we are able to make a prediction without relying on free parameters.”

Secondly, it outlines how the emerging field of 2D materials and heterostructures may contribute to the understanding of photo-induced superconductivity (also see answers to previous questions). Due to the versatile experimental control in these systems, this is a promising and novel route for the field.

We would like to thank the referee once more for giving us the opportunity to clarify this point.

In summary, it is my opinion that a closer and more realistic connection between theory and ultrafast spectroscopy experiments is needed to make progress at this stage of the field, given past reports and controversies. While such theory-experiment effort is certainly appropriate for Nature journals, it can be guided by simple models like the present one analyzed in more technical publications.

Our response: We would like to thank the referee for their assessment that the theory-experiment effort as suggested in our paper is appropriate for publication in nature journals. We hope to have convinced the referee that our work indeed makes a novel and substantial contribution to the field of photo-induced superconductivity and provides guidance into a new direction.

In summary, the key advancement of our paper is the microscopic nature of the mechanism we propose that is applicable to a wide range of material classes rather than being tailored to one specific experiment. As such it enables genuine predictions about new experiments without relying on phenomenologically introduced parameters that need to be fitted. Building on this we propose 2D heterostructures as a novel test ground for light-induced superconductivity,

providing a prediction for a concrete heterostructure for which calculations under experimental conditions have been performed.

The additions made to the paper following the points discussed by the referee have undoubtedly strengthened the paper and enhanced its clarity. We would therefore once more like to thank the referee for their assessment of our work and hope that together with the adjustments made it is now found suitable for publication in Nature Communications.

Reviewer #2 (Remarks to the Author):

In this manuscript, Eckhardt et al. proposed a mechanism of light enhanced superconductivity. A phonon that induces an electronic interband transition mediates an intraband attraction between the electrons. Since this process is proportional to the phonon occupation number, the authors propose that photo excitation of the phonon enhances this attraction. This effect is also resonantly enhanced when the phonon frequency approaches the interband threshold. The authors applied it to the sawtooth chain model and to a realistic graphene-SrTiO₃ heterostructure. This topic is of substantial interest. I would recommend its publication given that the following questions and issues are addressed:

Our response: We thank the referee for their precise recapitulation of our work and their recommendation for publication in Nature Communications given that their questions are addressed. Indeed they have raised important and interesting points to which we answer in detail below.

1, If one goes directly from the second term in Eq. 5 to Eq. 6, the occupation factor in Eq. 6 should be $b^{b+1/2}$ instead of b^b , similar to Ref. 14. This means that the resonance behavior should occur in equilibrium too. This is not the case found by the authors, as shown in Fig. 1b and discussed in the text. Why?

Our response: We thank the Reviewer for their insightful comment.

They are indeed correct that in principle one may expect such an equilibrium enhancement only from considering the second term in Eq.(5) of our manuscript. However, in the ground state and at resonance, this term precisely cancels with a contribution from the third term in Eq.(5) which is why we have not included the 1/2 in Eq.(6) (we commented on this in passing underneath Eq.(5) in the previous version of the paper).

We have added a clarification of this point underneath Eq.(6) that reads

“Eq. (6) shows that driving the bosons out of equilibrium will enhance this attraction transiently leading to a resonantly enhanced interaction. We note here that the resonance coming from the second and third term in Eq. (5), cancels in the ground state which is why the resonance is absent at equilibrium.”

2, In the estimates for the graphene-SrTiO₃ device, how large is the interband optical transition threshold. Is it above or below the surface phonon frequency (16.1 meV) of SrTiO₃? Above Eq. 32, it is said that the chemical potential is -15 meV, does it mean that the interband threshold is 30 meV?

Our response: We thank the referee for their question. The interband threshold can be visualized from Fig. 1(c) and actually equals half the band gap of graphene-hBN heterostructure + the chemical potential = 7 meV + 15 meV = 22 meV. This choice was taken so that the surface phonon polariton frequency is comfortably away from the bottom of the upper band and does not cause direct dipole transitions once excited. To clarify this point we have added close to the section that specifies the chemical potential the sentence:

“This choice of chemical potential leads to a finite carrier density in the lower band and corresponds to an interband optical transition threshold of 22 meV, preventing any direct dipole transitions due to the 16.1 meV surface phonon polariton oscillations.”

3, The SrTiO₃ phonon has a large linewidth, giving a large energy uncertainty. This probably means that the effective detuning between the phonon frequency and the interband threshold can never be very small. How does this fact change the estimations?

The referee makes a very good point here. It is indeed the case that the frequency of the boson mediating the interaction needs to be red-detuned with respect to the transition threshold by more than its linewidth.

We find in the literature that the linewidth of the phonon in SrTiO₃ that we consider is around ~ 10% (Phys. Rev. 174, 613 (1968)). Since we set the chemical potential such that it is red-detuned from the transition threshold by 6meV, this does not change our current estimate.

It is, however, an important point worth mentioning. We have therefore added a sentence to the methods sections which reads

“To prevent heating from direct dipole transitions, the surface phonon polariton must be red-detuned from the optical transition threshold by more than the phonon's linewidth, which in SrTiO₃ is around ~10%.”⁷⁴

Where the reference 74 in the updated version of the manuscript is the paper mentioned above. We thank the referee for drawing our attention to this important issue.

4, If the SrTiO₃ surface phonon frequency is above the interband threshold of graphene, the electrons in graphene are heated by the pump too. This may cause a serious heating given the much lower heat capacity of electrons. Is this an issue?

Our response: This is a very important point. Indeed, if the frequency is above the interband threshold, the pump will excite carriers into the upper band causing substantial heating which may be detrimental to superconductivity. This is why the chemical potential should always be

adjusted such that the frequency of the surface phonon polariton is red-detuned from this threshold. This coincides with the regime where we find an induced attraction for the electrons. If the frequency is red-detuned from the transition by more than the linewidth, as pointed out by the referee before, direct transitions are energetically forbidden. This is the case in our proposed experimental setup and therefore electron heating is expected not to be a major issue.

In order to clarify this point we added a sentence to the main part when describing how we estimate T_c for the concrete experimental setup we propose that reads

“The chemical potential is set such that the frequency of the surface mode is red-detuned from the inter-band transition by more than the linewidth, preventing direct excitations of electrons into the upper band due to the drive.”

We thank the referee for letting us highlight this important point which undoubtedly improved the clarity of the paper.

5, Typo in Fig. 2b: “Eq. 11” should probably be “Eq. 12

Our response: Indeed – the typo has been fixed.

We thank the reviewer once more for their concrete and detailed feedback to our paper, the inclusion of which in the updated version of the manuscript has certainly clarified important points. We hope the new version of the paper is now found suitable for publication in Nature Communications as is.

List of changes:

- Added data and code availability statements with links to the relevant repositories.
- Change the last sentence of the abstract which now reads:
“We discuss two-dimensional heterostructures as a potential test ground for light-induced superconductivity concretely proposing a setup consisting of a graphene-hBN-SrTiO₃ heterostructure, for which we estimate a superconducting T_c that may be achieved upon driving the system.”
- Change the beginning of the second paragraph of the introduction that now reads:
“In the context of equilibrium superconductivity in doped SrTiO₃,^{34–37} the importance of local interband phonon coupling has previously been discussed. In this paper, however, we highlight the non-equilibrium properties of this model that to the best of the authors’ knowledge have not been explored to date in the context of driven superconductivity. This can potentially elucidate the microscopic mechanism behind photo-induced superconductivity.”
- Change the beginning of the third paragraph of the introduction that now reads:
“Finally we put forward two-dimensional heterostructures as a test-bed for our theory. In particular, we propose to couple two-dimensional materials to the surface phonon polaritons of a substrate. Previous materials, in which light-induced superconductivity

has been observed like cuprates,³⁸ κ -salts¹³ and K3C60,^{10–12} are strongly correlated hindering the formulation of simple theories as well as providing limited experimental control handles, making the verification of proposed mechanisms difficult. In contrast, two-dimensional (van der Waals) materials as well as surface phonon polaritons offer precise control potentially enabling direct tests of concrete microscopic theory proposals for light-induced superconductivity. As a concrete and practically realizable example, we explore a heterostructure consisting of graphene aligned with hexagonal boron nitride ...”

- Adding the reference: *D. Fausti et.al., Science 331, 189–191 (2011)*
- Clarified Eq.(6) with the statement
“Eq. (6) shows that driving the bosons out of equilibrium will enhance this attraction transiently leading to a resonantly enhanced interaction. We note here that the resonance coming from the second and third term in Eq. (5), cancels in the ground state which is why the resonance is absent at equilibrium. ”
- *Fixed typo in Fig.2b.)*
- Added a clarification of the tuning of the chemical potential in our experimental prediction that reads:
“The chemical potential is set such that the frequency of the surface mode is red-detuned from the inter-band transition by more than the linewidth, preventing direct excitations of electrons into the upper band due to the drive.”
- We now predict further decisive experimental evidence the measurement of which would strongly suggest the applicability of our mechanism by writing underneath our estimate for a transient T_c
“A distinctive test of our theory may be performed by further reducing the chemical potential in graphene which, according to the proposed mechanism, is expected to decrease T_c despite resulting in a larger density of states at the Fermi-surface.”
- We clarify that our prediction has no fitting parameters in the discussion part stating:
“Owing to the microscopic nature of our mechanism, we are able to make a prediction without relying on free parameters.”
- In the methods section concerning our experimental prediction, we clarified the interband threshold set by our choice of the chemical potential as well as the importance to consider the linewidth of the inter-band boson when computing the red shift. We added the statement:
“This choice of chemical potential leads to a finite carrier density in the lower band and corresponds to an interband optical transition threshold of 22 meV, preventing any direct dipole transitions due to the 16.1 meV surface phonon polariton oscillations. To prevent heating from direct dipole transitions, the surface phonon polariton must be red-detuned from the optical transition threshold by more than the phonon's linewidth, which in SrTiO₃ is around $\sim 10\%$.⁷⁴”
- We added a new reference 74, *Phys. Rev. 174, 613-623 (1968)*.
- We updated reference 12 to the now available published version.
- We consistently removed vector arrows throughout the paper except for one paragraph in the methods section where they are used to distinguish quasi-momenta from composite indices as now stated in that paragraph.

- We updated the notation in Eq.(26) in the methods section for clarity adding a statement about the notation below.
- In the first paragraph of the methods section on the Schrieffer-Wolf transformation, in Eq.(18) and Eq.(20) and the text below Eq.(20), we fixed typos related to notation, consistently denoting the bosonic frequency by a capital omega and the energy gap by Delta E.

REVIEWERS' COMMENTS

Reviewer #2 (Remarks to the Author):

My questions are addressed in the revised manuscript. I recommend its publication.